# Surrogate modeling for time-dependent reliability analysis of robotic manipulator trajectories

Keenjhar Ayoob[1], Hassan Elahi[1]*, Tayyab Zafar[1]*, Amir Hamza[1], Zhonglai Wang[2,3]

**1** National University of Sciences and Technology (NUST), sector H-12, Islamabad, Pakistan, **2** University of Electronic Sciences and Technology of China, Chengdu, Sichuan, P.R.China, **3** Institute of Electronics and Information Industry Technology of Kash, Kaush, Xinjiang, P.R. China

* hassan.elahi@ceme.nust.edu.pk (HE); tayyab.zafar@ceme.nust.edu.pk (TZ)

## Abstract

The kinematic reliability analysis of robotic manipulators is crucial due to uncertainties such as joint variations, manufacturing tolerances, and external disturbances. Traditional methods often rely on analytical techniques that struggle with nonlinear performance functions and fail to account for trajectory-based reliability. To overcome these limitations, this paper proposes a novel surrogate model-based approach using Kriging to estimate the reliability of robotic manipulator kinematics while considering end-effector trajectories. The methodology begins with building an initial Kriging surrogate model to analyze reliability, effectively capturing how input uncertainties influence trajectory accuracy. This model is then refined through statistical sampling techniques, ensuring an efficient evaluation of manipulator performance against specified tolerances. The approach reduces computational complexity while maintaining prediction accuracy. Compared to Monte Carlo Simulation (MCS), the proposed Kriging-based method reduces the number of function evaluations by over 98%, achieving comparable reliability predictions with significantly fewer function calls, and enhancing efficiency in kinematic reliability analysis. The proposed method is validated on two 6-DOF industrial robots, including the UR5, demonstrating improved computational efficiency and accuracy. This work has practical applications in manufacturing and healthcare, where enhanced kinematic reliability leads to greater operational efficiency and safety.

## 1. Introduction

Precision in robotics and kinematics plays a crucial role in enhancing the accuracy and performance of robotic manipulators, particularly in applications where accurate performance is essential. Uncertainties such as joint tolerances, manufacturing inaccuracies, and external disturbances can significantly affect the trajectory accuracy and overall performance of these systems [1]. Achieving high levels of kinematic precision ensures that robotic systems perform complex operations with minimal error

**Data availability statement:** The data underlying the results presented in this study are incorporated in the paper.

**Funding:** The author(s) received no specific funding for this work.

**Competing interests:** The authors have declared that no competing interests exist.

and can effectively manage uncertainties, enabling them to function reliably in critical applications such as manufacturing and healthcare [2]. Trajectory accuracy and reliable control mechanisms are even more crucial when robots perform sensitive tasks in dynamic or unpredictable environments [3]. Therefore, improving precision by minimizing actuation errors and extending the manipulator's range of movement contributes to a more reliable system [4,5]. and unlike traditional neural networks, which primarily estimate global errors, [6] our proposed Kriging meta-model can calculate the error of each sample separately, ensuring a more precise reliability assessment.

Despite this, the problem lies in maintaining consistent accuracy over prolonged operation. Manipulator precision may weaken because of mechanical wear, corrosive environments, and various operating factors. Yet, the performance of key components, for example, couplings, bearings, and actuators, can weaken because of degradation in their precision [7]. External conditions like temperature fluctuations, humidity, dust, and chemicals can also compromise the quality of the structure [4]. Consistent maintenance and calibration of control algorithms are crucial to enhance the effectiveness of robotic manipulators [8].

Several methods have been used for the analysis of kinematic reliability under uncertainty. Some of these methods are saddle point approximation, moment-based approaches, and other mathematical methods that offer solutions to difficult reliability problems—for instance, Zhao et al. [9] presented a moment-based approach that used the theory of Lie groups and series expansion to simplify the limit state function and improve the accuracy of reliability assessment by chi-square approximation. Zhao and Hong. [10] proposed an analytical method to assess the kinematic sensitivity of reliability using uncertainty analysis and non-central chi-square approximation. Zhang et al. [11] suggested a moment-based method which was based on mixed degree cubature formulas and vine copula functions to manage uncertainty in complex mechanical systems. Zhang et al [12] proposed the algorithm of saddle point approximation for kinematic reliability analysis of robotic manipulators and obtained promising accuracy and efficiency to predict the reliability of the end-effector's position. Cui et al. [13] implemented the Monte Carlo simulation algorithm to detect the kinematic accuracy and reliability analysis of a 3-DOF parallel robot manipulator and achieved a theoretical basis for design optimization and error compensation.

Wu and Rao [14] focused on tolerance allocation methods, employing interval analysis to predict performance errors. Zhang et al. [15] Introduced a kinematic trajectory accuracy and reliability analysis method for industrial robots, combining the sparse grid technique, saddle point approximation method, and copula functions to assess intercorrelations in positioning errors and enhance reliability evaluation. Huang et al. [3] expanded on moment-based methods for trajectory accuracy in conditions of random and interval uncertainties. Zhang et al. [16] proposed an efficient multi-failure mode reliability method using intelligent directional search. This study uses cutting-edge optimization strategies for well-known industrial robots. These models, widely cited in research and industry, are used to assess solution feasibility and refine robot trajectory planning and control based on Denavit-Hartenberg parameters [12]. Hawchar et al. [17] Introduced the Polynomial Chaos Expansion (PCE)

algorithm for time-variant reliability analysis and obtained effective results in modeling high-dimensional, time-dependent problems with non-stationary and non-Gaussian processes. Chen et al. [18] introduced an Improved Approximate Integration Method (IAIM) to improve accuracy in solving nonlinear reliability issues, which supports better precision in robot control and planning. The analytical methods demonstrate excellent potential through their use of advanced computational systems which remain complex to handle.

Monte Carlo Simulation (MCS) functions as the primary solution to simulation-based methods because it provides flexible alternatives to the restrictions of analytical techniques. Robotics engineers use Monte Carlo Simulation as their primary tool when analyzing reliability because it helps identify failure probabilities by applying numerous simulated robotic configurations. Cai et al. [19] used MCS in a time-dependent reliability analysis for a 6-DOF serial-parallel precision positioning system. The study conducted by Ali et al. [20] employed MCS for reliability evaluation within robotic manipulators that utilized 3D printing while Yun and Li [21] applied MCS for parallel robot analysis. Hu et al. [22] combined MCS with Kriging surrogates for improved time-dependent reliability analysis of structural systems. A combined method of dynamic adaptive enhanced simulation with support vector regression (SVR) was developed by Luo et al. [23] for better structural reliability analysis. In addition to these methods, several other techniques have been proposed to address the time-dependent reliability of robotic manipulators. The end-effector accuracy and reliability improvements of robotic manipulators were achieved through probabilistic modeling techniques described by Rao et al. [24]. The combination of Bayesian reinforcement learning with MCS by Zhou et al. [25] allowed researchers to conduct sequential experiments for decreasing uncertainty in complex systems.

Jiang et al. [26] and Hu and Mahadevan [27] offered single-loop Kriging surrogate modeling, aiming to enhance efficiency in time-dependent reliability analysis. Although robust, MCS-based methods are computationally expensive due to the large number of required simulations, limiting their use in real-time applications.

To address the computational inefficiency of simulation-based methods, surrogate models like Kriging have been introduced as alternatives. These models approximate the relationship between input uncertainties and output reliability without the need for exhaustive simulations. Wang et al. [28] developed an adaptive Kriging model with n-hypersphere rings (AK-HRn) for efficient reliability analysis of complex problems. Liu et al. [29] combined evidence theory with convolutional neural networks (CNNs) to improve reliability analysis accuracy by classifying uncertainties in engineering structures. Dang et al. [30] demonstrated Bayesian active learning methods in Kriging models (PBALC1, PBALC2, PBALC3), showing improved efficiency in small failure probability assessment. Zhuang et al. [31] proposed an envelope method incorporating dynamic factors like material wear for kinematic reliability in motion mechanisms. Hong et al. [32] introduced a portfolio allocation strategy inspired by the multi-armed bandit approach to enhance the selection of Kriging-based learning functions for structural reliability analysis, demonstrating its effectiveness through various numerical and engineering applications. Qian et al. [33] introduced a double-loop Kriging model for time-variant reliability in industrial robots, while Luo et al. [34] coupled Kriging with the conjugate first-order reliability method (AK-CFORM) for structural assessments. Zafar et al. [35] also explored Kriging for time-dependent reliability, and Shi et al. [36] presented an adaptive multiple-Kriging model for estimating time-dependent failure probability, showing increased accuracy over fixed regression models.

Existing literature on kinematic reliability analysis has primarily focused on time-independent methods, often overlooking dynamic factors that influence robotic manipulators' trajectory prediction. While surrogate models have improved efficiency, they have not effectively addressed the complexities of time-dependent reliability, particularly for highly nonlinear robotic systems.

A new simulation-based approach using surrogate modeling handles the issues in the kinetic reliability analysis of robotic manipulators that show time-dependent behavior. The method uses time-dependent reliability analysis and Kriging-based surrogate modeling to provide precise results while minimizing computation expense. Monte Carlo Simulation (MCS) serves as a validation to test the method before its application on 6-DOF industrial robots, resulting in effective

kinematic structure analysis. The technique proves superior to MRSM and existing prediction methods while providing real-time efficiency, which makes it a practical solution for robotic implementations.

The paper is structured as follows: Section 2 discusses the problem formulation. The proposed methodology is explained in Section 3. Section 4 discusses case studies for validation purposes. Section 5 is the discussion section. The paper concludes in section 6.

## 2. Problem formulation

### 2.1. Time-dependent reliability analysis

Time-dependent reliability in 6-DOF robotic arms considers trajectory evolution, incorporating uncertainties that cause deviations from the designated path $G_D$ to the practical path $G_P$. The Denavit-Hartenberg (DH) parameters provide the joint angles, link lengths, and relative link positions [37]. The positional deviation of the end-effector can be expressed as:

$$\Delta E = \left(\Delta \Phi^T, \Delta Q^T\right)$$

(1)

Where: $\Delta \Phi^T$ represents end-effector pose error, and $\Delta Q^T$ denotes joint position error. These are key for assessing manipulator reliability. The following sections detail their quantification and time-dependent reliability evaluation. The time-dependent reliability of a robotic arm is calculated as the probability that its positional deviation remains within a safety margin from $t_{int}$ to $t_{term}$.

$$R_T\left(t_{int}, t_{term}\right) = Pr(E), \; E : \; \forall t \in [t_{int}, t_{term}], \| \Delta Q(t) \|^2 \leq s^2$$

(2)

Here, $\Pr\{\cdot\}$ signifies the likelihood, where "s" denotes the radius of the defined spherical safety boundary, and the subscript "R" is indicative of reliability. In this scenario, the time-independent probability of failure is as given by:

$$P_F(t) = 1 - P_R(t)$$

(3)

The subscript "F" stands for failure, and the subscript R represents reliability. Imagine a robotic arm tracing a specified path (illustrated as the solid red line in Fig 1. Various uncertainties, including sensor errors, inaccuracies in modeling, or external influences, can cause the real path taken by the robot to differ from the intended one. We will explore three distinct paths:

Path 1: Observes a breach of the designated safety limit during the period from $t_1$ to $t_2$.

Path 2: Consistently stays within the defined safety margin throughout the duration from $t_{int}$ to $t_{term}$.

Path 3: Witnesses a breach within the timeframe from $t_1$ and $t_2$. It is important to note that should any segment of a path surpass the predefined positional tolerance, the motion for the entire duration is deemed unsuccessful. Mathematical Formulation of this is as given in equation 3:

$$P_R\left(t_{int}, t_{term}\right) = Pr\{E_d(Y, t) \leq s^2, \forall t \in [t_{int}, t_{term}]\}$$

(4)

where: $E_d(Y, t)$ represents the pose error at the time $t$ and $s$ is the allowable specification (safe boundary). The symbol $\forall$ stands for "for all." Similarly, the failure probability within a time interval from $t_{int}$ to $t_{term}$ is expressed as:

$$P_F\left(t_{int}, t_{term}\right) = 1 - P_R(t_{int}, t_{term})$$

(5)

The time-dependent positional reliability is the probability that the maximum positional error throughout the motion stays within the safe limits. This concept allows us to simplify Equation (2) into Equation (3):

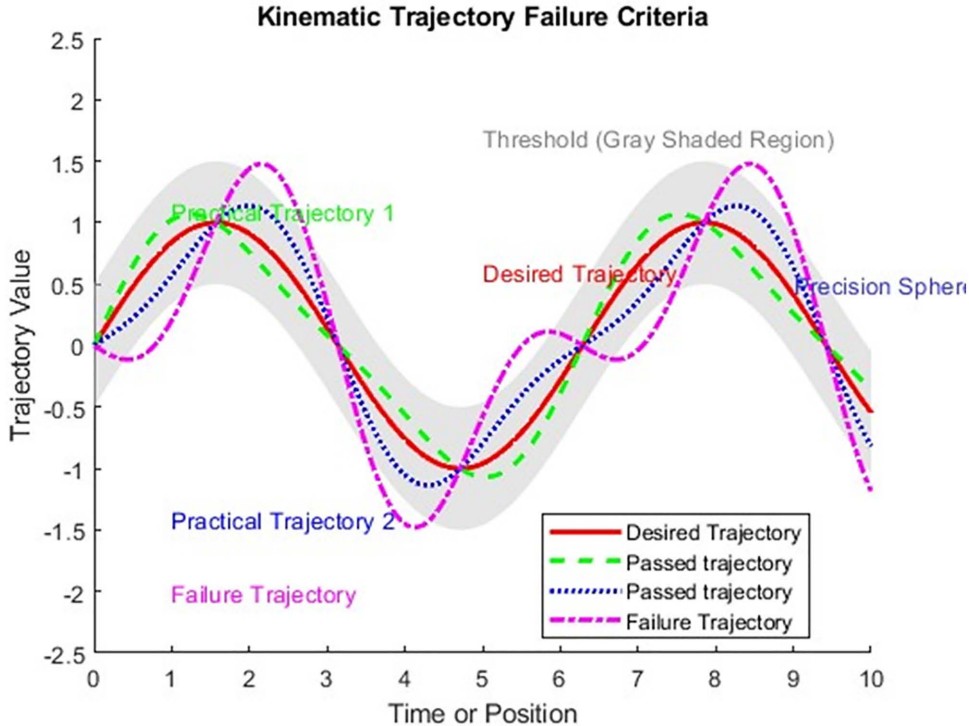

**Fig 1. Comparing Technical Parameters to Assess the Reliability of the Placement of Robotic Manipulator.**

$$P_R = P_r\{E_{max} \leq s^2,\ E_{max} = max(E_d\,(Y, t_1)\,, E_d\,(Y, t_2)\,, \ldots, E_d(Y, t_{term}))$$

(6)

The robotic arm reliability estimation incorporates the combination of a polynomial error model with a non-central chi-square method to avoid direct measurement techniques. $R_T(t_{int}, t_{term})$ Calculates the likelihood of endpoint position maintenance within safety parameters [9] through statistical performance evaluation methods [12,38].

## 2.2. Kinematic modeling

Accessing ($E_D\,(Y, t)$) involves estimating the maximum positional offset and system reliability to enhance robotic manipulator accuracy, achieved through forward kinematics where the pose is determined by multiplying transformation matrices.

$$G_D = \Pi_{\sqsupset=1}^{6}\ T_{i-1}^{i}$$

(7)

The transformation matrix $T_{i-1}^{i}$ maps the $(i-1)-$th to $i-th$ joint using modified DH parameters, combining rotations and translations. It is crucial for kinematic modeling, trajectory planning, and motion control.

$$T_{i-1}^{i}\ =\ R_z\,(\theta_i)\,.\ T_x\,(a_i)\,.T_z\,(d_i)\,.\ R_x(\propto_i)$$

(8)

This compact representation enhances robotic manipulators' efficiency, precision, and speed, ensuring smooth motion and accurate end-effector positioning. The resultant transformation matrix $T_{i-1}^{i}$ is formed by combining these transformations.

$$T_{i-1}^i = \begin{bmatrix} cos(\theta_i) & -sin(\theta_i) & 0 & a_i \\ sin(\theta_i)cos(\propto_i) & cos(\theta_i)cos(\propto_i) & -sin(\propto_i) & -sin(\propto_i)\,d_i \\ sin(\theta_i)sin(\propto_i) & cos(\theta_i)sin(\propto_i) & cos(\propto_i) & cos(\propto_i)\,d_i \\ 0 & 0 & 0 & 1 \end{bmatrix} \qquad (9)$$

Overall Transformation for a manipulator with 6 links, the overall transformation matrix $G_D$ from the base to the end effector is obtained by multiplying the individual transformation matrices, we can express the overall transformation $G_D$ as:

$$G_D = \Pi_{i=1}^6 \begin{bmatrix} cos(\theta_i) & -sin(\theta_i) & 0 & a_i \\ sin(\theta_i)cos(\propto_i) & cos(\theta_i)cos(\propto_i) & -sin(\propto_i) & -sin(\propto_i)\,d_i \\ sin(\theta_i)sin(\propto_i) & cos(\theta_i)sin(\propto_i) & cos(\propto_i) & cos(\propto_i)\,d_i \\ 0 & 0 & 0 & 1 \end{bmatrix} \qquad (10)$$

This takes place in such a manner that it may generally be useful in determining the pose of the manipulator involving a position offset, as well as other system parameters and reliability checks. The new DH parameters, as in Fig 2, adapted in this paper, provide kinematic analysis with a well-defined structure that provides methods of categorization of robotic manipulators with a high degree of accuracy in modeling them.

## 2.3. Mathematical formulation

The concept of reliability analysis across time, which was first proposed in the third equation, may be modified to more precisely represent dynamic correctness. Availability assessment involves predicting the probability that the system will function within a satisfactory limit for a given period. This may be stated as follows:

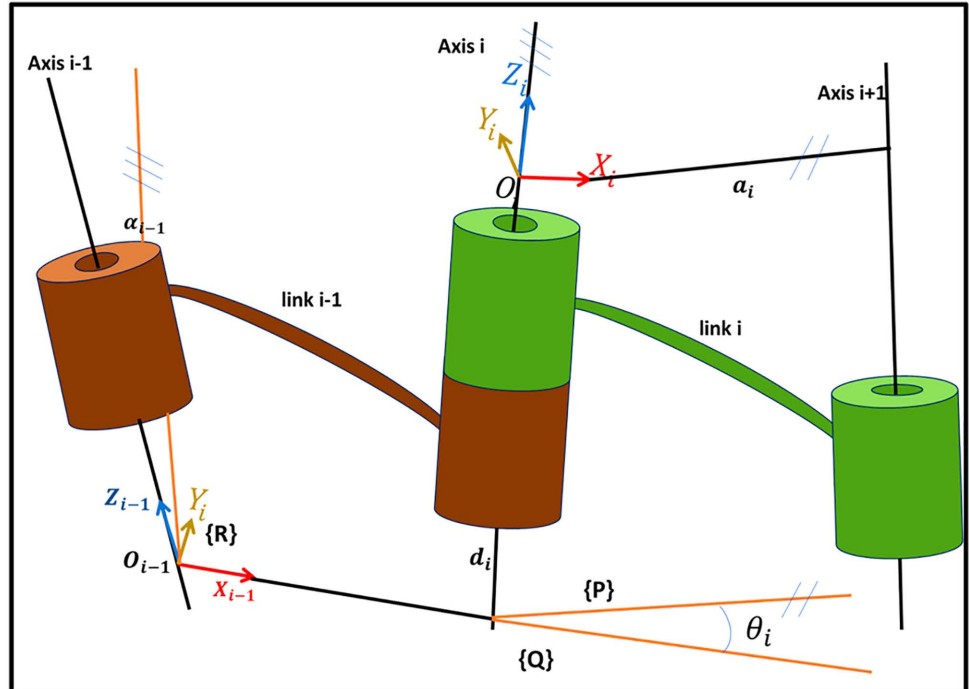

**Fig 2. DH- parameter.**

$$P_R(t_{int}, t_{term}) = P_R\{E_{max} < s^2, E_{max} = max(E(Y, t_1), E(Y, t_2), \cdots, E(Y, t_{term}))\} \tag{11}$$

Equation (11) defines reliability $P_R$ over $[t_{int}, t_{term}]$ based on the maximum positional error $E_{max}$. The system remains accurate if $E_{max}$ stays below the threshold $S^2$. Extending this in dynamic accuracy offers a quantitative measure of the manipulator's precision and reliability over time.

Key information from the practical end-effector position $G_P$ is extracted by analyzing positional deviations in $x$, $y$, and $z$, orientation angles (rotational variations), and trajectory quality (pass/fail rates). These factors provide a clear assessment of manipulator performance, aiding decisions to enhance reliability and productivity. To account for uncertainty, randomness is introduced in joint angles and link parameters, with $\theta_i$ following a uniform distribution centered around its nominal mean $\bar{\theta}_i$.

$$\theta \sim U(\theta_{min}, \theta_{max}) \tag{12}$$

Similarly, the link offsets $d_i$ and link lengths $a_i$ are subject to uncertainty and modeled using normal distributions. For each link offset and link length, we introduce variability around their nominal values with a standard deviation $Std_d$ for $d_i$ and $Std_a$ for $a_i$ as:

$$d_{i,uncertain} \sim \mathcal{N}(\bar{d}_i, \; Std_d), \; a_{i,uncertain} \sim \mathcal{N}(\bar{a}_i, \; Std_a) \tag{13}$$

The normal distribution $N(\mu, \sigma)$ defines variations in manipulator parameters. The FK model computes the 3D end-effector position over multiple periods, with each period yielding a distinct position. Multiplying the transformation matrices gives the final position.

$$G_D = G_0^1 \cdot G_1^2 \cdot G_2^3 \cdot G_3^4 \tag{14}$$

Theoretical and stochastic trajectories define ideal and uncertain end-effector paths. $G_P$ adjusts for uncertainties, refining estimation, and is expressed as follows:

$$G_P = \Pi_{i=1}^6 \begin{bmatrix} \cos(\theta_i + \Delta\theta_i) & -\sin(\theta_i + \Delta\theta_i) & 0 & a_i + \Delta a_i \\ \sin(\theta_i + \Delta\theta_i)\cos(\alpha_i + \Delta\alpha_i) & \cos(\theta_i + \Delta\theta_i)\cos(\alpha_i + \Delta\alpha_i) & -\sin(\alpha_i + \Delta\alpha_i) & -\sin(\alpha_i + \Delta\alpha_i)(d_i + \Delta d_i) \\ \sin(\theta_i + \Delta\theta_i)\sin(\alpha_i + \Delta\alpha_i) & \cos(\theta_i + \Delta\theta_i)\sin(\alpha_i + \Delta\alpha_i) & \cos(\alpha_i + \Delta\alpha_i) & \cos(\alpha_i + \Delta\alpha_i)(d_i + \Delta d_i) \\ 0 & 0 & 0 & 1 \end{bmatrix} \tag{15}$$

## 3. Proposed methodology

The proposed methodology follows a structured approach, integrating Kriging surrogate modeling with Monte Carlo Simulation (MCS) for efficient reliability estimation of robotic systems. The Kriging model approximates forward kinematics, while MCS validates model accuracy and estimates failure probabilities. An adaptive sampling strategy refines the surrogate model, focusing on high-uncertainty regions to improve computational efficiency.

### 3.1. Kriging surrogate model

The input parameters for the robotic system are defined as: $X_{int} = \{\theta_i, \; d_i, \; a_i, \; \alpha_i\}$ Where $\theta_i$: Joint angles, $d_i$: Link offsets, $a_i$: Link lengths, $\alpha_i$: Twist angles. The end-effector position in 3D space, $\hat{Y}$, is computed using the Denavit-Hartenberg (DH) kinematic model: $\hat{Y} = f(X_{int})$. For $n = 10$ initial samples, the output matrix $\hat{Y}$ is:

$$\hat{Y} = \begin{bmatrix} x_1 & y_1 & z_1 \\ x_2 & y_2 & z_2 \\ \vdots & \vdots & \vdots \\ x_{10} & y_{10} & z_{10} \end{bmatrix} \tag{15}$$

The output matrix $\hat{Y}$ in  is computed using forward kinematics from a set of 10 input samples. These samples are generated by perturbing the Denavit–Hartenberg (DH) parameters based on realistic uncertainty models. Joint angles, $\theta_1$ and $\theta_5$ are calculated from uniform distributions, while link lengths and offsets follow normal distributions as shown in the following pseudocode 1. It provides an even distribution of points throughout the space of given inputs. Forward kinematics (using the Peter Corke Robotics Toolbox) compute corresponding end-effector positions. The process also makes the error rate more precise by adjusting the model in problem areas.

```
Pseudocode 1: for data initial samples Generation:
Initial Sample Generation
for i = 1 :N_samples
  θ₁ = unifrnd(90, 450);
  θ₅ = unifrnd(-90, 270);
  d = normrnd(d_nominal, σ_d);
  a = normrnd(a_nominal, σ_a);
  α = fixed or DH-defined;
  T = Forward_Kinematics(θ, d, a, α);
  Store T and DH parameters;
End
```

### 3.2. Kriging model formulation

The Kriging model $\hat{G}(N)$ at a sample point $N$ is expressed as:

$$\hat{G}(N) = D_a(N)\delta_a + \omega_u P(N) \tag{17}$$

Where, $D_a(N)$: Regression function matrix (e.g., polynomial basis functions), $\delta_a$: Regression coefficients (estimated via least squares), $\omega_u$: Gaussian process variance, $P(N)$: Zero-mean Gaussian process with unit variance. The coefficients $\delta_a$ are calculated using:

$$\delta_a(\phi_a) = \left(\nabla_u Y^{-1} \nabla_u\right)^{-1} \nabla_u Y^{-1} G_n \tag{18}$$

Where, $\nabla_u$: Gradient matrix of regression functions, $G_n$: Observed outputs at training samples, the variance $\omega_u^2$ is computed as:

$$\omega_u^2(\phi_u) = \frac{1}{n}\left(G_a - \nabla_a\delta_u\right)Y^{-1}(G_n - \nabla_a\delta_u) \tag{19}$$

$Y$: Covariance matrix of outputs. The correlation parameters $\phi_u$ are optimized via Maximum Likelihood Estimation (MLE):

$$\phi_u^{MLE} = min(\omega_u^2|Y|) \tag{20}$$

For a new sample $N_e$, the Kriging model predicts:
  Mean:

$$\hat{\alpha}_u(N_e) = D_u(N_e)\delta_u + h_u(N_e)Y^{-1}(G_n - \nabla_u\delta_u) \tag{21}$$

Variance:

$$\hat{\omega}_u(N_e) = \omega_u^2(1 - (G_n(N_e))w(N_e)) \tag{22}$$

In equation 22 $h_u$ Correlation function for $N_e$ and $w(N_e)$ Correlation vector at $N_e$. It is particularly useful that the Kriging surrogate model can generate estimates of the end-effector position and a sample-wise calculation of how uncertain each prediction is. The model, for every input sample, returns the predicted position as well as a Mean Squared Error reading that shows how confident it was in that prediction. This makes the model particularly powerful for reliability analysis, as it allows error-sensitive decisions to be made at each test point. The adaptive sampling loop counts on sample-wise MSE values to direct additional training to the regions that are still uncertain for the model. This built-in capability to quantify and localize error propagation enhances both model transparency and reliability assessment accuracy.

### 3.3. Adaptive sampling strategy

The error function $U_u(N_e)$ evaluates prediction confidence:

$$U_u(N_e) = \frac{\hat{\alpha}_u(N_e)}{\hat{\omega}_u(N_e)}$$

(23)

If $U_u(Ne) > \tau$ (e.g., $\tau = 2$), the sample is accepted. The error reduction function (ERF) is computed as:

$$ERF = \frac{erf(\hat{\omega}_u(N_e))}{1.892\sqrt{2}}$$

(24)

and the Mean Squared Error (MSE)-based Efficient Function (EFF) is given by:

$$EFF = \frac{\hat{\omega}_u(N_e)}{1 + ef}$$

(25)

### 3.4. Adaptive Kriging-Based Reliability Estimation

The proposed methodology iteratively refines the Kriging surrogate model using adaptive sampling and convergence checks. The following pseudocode 2 outlines the process. This iterative process ensures that the model's accuracy improves in areas where it was previously uncertain. The following are the steps of our proposed methodology.

Pseudocode: 2 of the proposed methodology

```
Algorithm: Adaptive Kriging-Based Reliability Estimation
1: Procedure Adaptive-Kriging the g (X_int, Ŷ)
2:     Initialize X_int, compute Ŷ
3:     Train the Kriging model using Equations (17-20)
4:     repeat
5:         Predict outputs for new samples using Equations (21-22)
6:         Evaluate errors using, U, ERF, EFF (Equations 23-25)
7:         Classify trajectories into Group 1 and Group 2
8:         Compute P_F and E_r
9:         if E_r ≥ 5% then
10:             Select high-uncertainty samples for refinement
11:             Update the Kriging model with refined samples
12:         end if
13:     until E_r < 5%
14:     return trained Kriging model and final reliability estimates
15: end procedure
```

Validation Framework: Trajectories with errors greater than 2 mm are classified into Group 1 (failing) or Group 2 (passing). Additional thresholds (0.5 mm, 1.2 mm, 1.5 mm) refine trajectory classification. Failure probability is computed as:

$$P_F = \frac{N_{F1} - N_F}{N_{F2} + N_F} \tag{26}$$

$$P_R = 1 - P_F \tag{27}$$

Where, $N_{F1}$: Failures in Group 1 and $N_{F2}$: Failures in Group 2. The error is calculated for validation purposes:

$$E_r = \frac{P_F(kriging) - P_F(MCS)}{P_F(MCS)} \times 100 \tag{28}$$

If $E_r$ < 5% The simulation terminates. Fig 3 shows that the iterative process ensures that the model's accuracy improves in areas where it was previously uncertain. The following are the steps of our proposed methodology.

**Step 1:** Uncertainty in joint angles and link parameters is modeled using uniform and normal distributions around their nominal values. These variations generate configurations, and the forward kinematics (FK) model computes the corresponding end-effector positions.

**Step 2:** The Kriging model is trained to predict the end-effector coordinates ($x$, $y$, $z$) based on Denavit-Hartenberg (DH) parameters. Ten input samples are generated, incorporating uncertainties in joint angles ($\theta$), link lengths ($a$), and

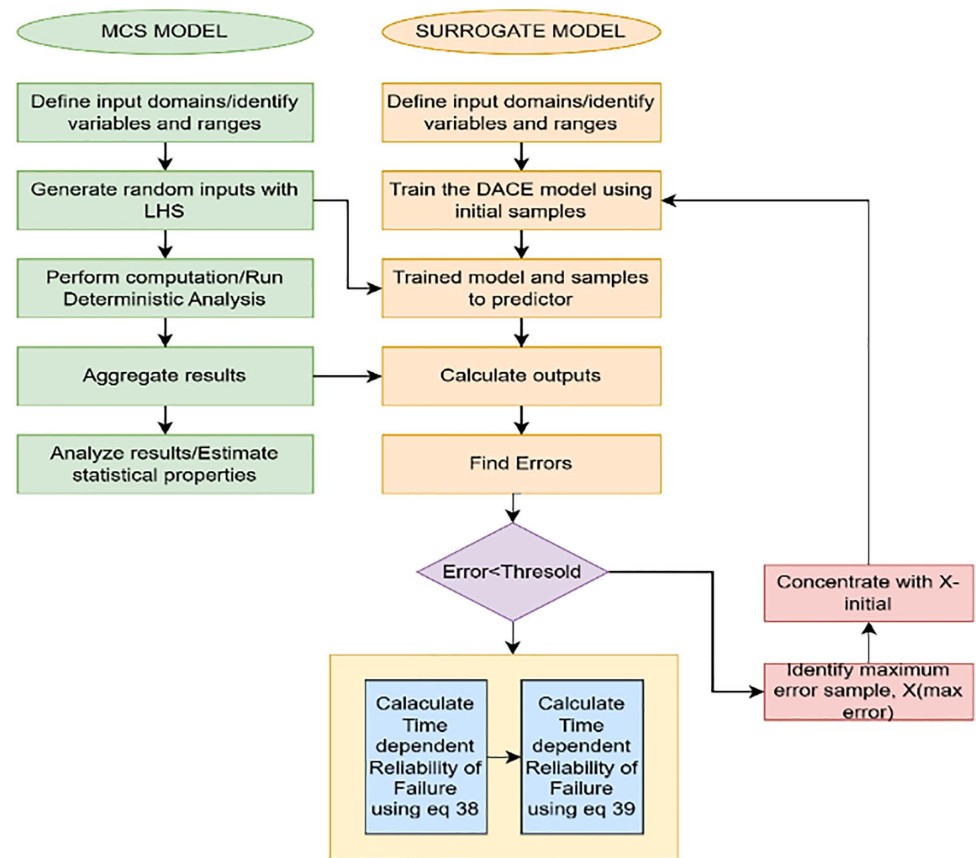

**Fig 3. Proposed algorithm.**

link offsets ($d$). After eliminating redundant columns, the reduced matrix ($10 \times 11$) is used with the forward kinematics function to compute corresponding 3D end-effector positions ($10 \times 3$). Separate Kriging models are then trained for each coordinate ($x$, $y$, $z$) using the input-output pairs, resulting in three models (M1, M2, and M3) that map the DH parameters to the end-effector positions.

 **Step 3:** The trained Kriging models predict end-effector positions ($x$, $y$, $z$) for a large set of new input configurations. The input matrix is reduced to relevant Denavit-Hartenberg parameters, and predictions are generated for each coordinate using the models M1, M2, and M3. The predictions are combined into a single matrix P, with Mean Squared Error ($MSE$) computed for each prediction. Key metrics—Uncertainty Index ($U$), Error Reduction Factor ($ERF$) and Efficiency Factor ($EFF$) are calculated to assess prediction reliability and guide adaptive sampling, focusing on high-error regions to refine the model further.

 **Step 4:** the accuracy of failure predictions for each trajectory is assessed by classifying them into passing ($P$) and failing ($F$)categories using error metrics ($U$, $ERF$, $EFF$) and a threshold of $2.5\ mm$. The number of passing ($NP$) and failing ($NF$) trajectories is counted, and the error is calculated as the difference between the predicted and validated failure counts, normalized by the validated failure count. The adaptive sampling process continues until the error for each trajectory is below the threshold, with termination occurring when the error meets the condition $\in \leq \theta$.

 **Step 5:** The Kriging model is refined by adding high-error data points, identified using error matrices ($U$, $ERF$, $EFF$), to the training set. The maximum error and its index are determined from the error matrix, and the corresponding Denavit-Hartenberg (DH) parameters and Forward Kinematics (FK) results are extracted. These high-error data points are then added to the training set, increasing its size by $1$ with each iteration. The process repeats until the error matrix reaches a predefined threshold, at which point the adaptive sampling process terminates.

## 4. Case studies and results

The proposed methods reduce function evaluations while maintaining accuracy. Kriging-based models significantly lower computational costs compared to MCS while ensuring reliable predictions. This approach enhances robotic performance with robust reliability estimation. Computations were performed in MATLAB 2023a on an Intel i5-6300U CPU with 8 GB RAM.

### 4.1. Reliability analysis of a 6-DOF Robotic Manipulator

In this section, the effectiveness and advantages of the proposed method are demonstrated using a robotic manipulator, with Monte Carlo Simulation (MCS) performed for comparison. Additionally, a comparison with previous works is conducted, highlighting its superior performance.

 The 6-DOF robotic manipulator's DH parameters are shown in Table 1 and Table 2 shows the drive designs for the first and fifth joints, which are actuated. There are predetermined angles for the second, third, fourth, and sixth joints. Fig 1 shows the three-dimensional path, which has been discretized into 45 points for reliability study. Table 3, which includes the means and standard deviations, details the uncertainties. In this study, three variant Kriging models, *Kriging* $+$ $U$, *Kriging* $+$ *EFF*, and *Kriging* $+$ *ERF* are proposed as the primary techniques for reliability estimation, Kriging is proposed because it is computationally efficient and, unlike other methods such as neural networks and MRSM, it not only predicts results but also performs error analysis for each sample. The performance of these Kriging models is compared with MRSM and PCE. To compute positional errors, generate samples for each random error variable, and determine time-dependent probabilities of positional reliability, Monte Carlo Simulation (MCS) is employed as a benchmark [39]. For reliability estimation from the initial to the last point, $P_R(t_{int}, t_{term}) = N_R N$ is proposed. where N is the overall sample size and $N_R$ is the subset of that size that falls inside the risk-free zone. Here, $N$ is set at $55 \times 10^4$. Based on probabilities for safe bounds ranging from $2\,mm$ with a difference of $0.5$ till $3.5 mm$, the time-interval trajectory reliability across $[t_{int}, t_{term}]$ is computed. The interval probabilities of trajectory reliability under four safe bounds are shown in Table 2, demonstrating that our technique outperforms others. Below is an example of $P_R\ (t_{int}, t_{term})$ under various safe limits; in cases with low

**Table 1. DH Parameters for the 6-DOF Robotic Manipulator.**

| Joint | $\alpha_i$ (degrees) | $d_i$ (mm) | $a_i$ (mm) | $\theta_i$ (degrees) |
|---|---|---|---|---|
| 1 | 90 | 0 | 0 | $\theta_1$ |
| 2 | 0 | $d_2$ | $a_2$ | 90 |
| 3 | 0 | 0 | $a_3$ | 0 |
| 4 | 0 | 0 | $a_4$ | −135 |
| 5 | −90 | 0 | 0 | $\theta_5$ |
| 6 | 0 | $d_6$ | 0 | 0 |

**Table 2. Driving Strategies.**

| Time Interval | $\theta_1$ (degrees) | $\theta_5$ (degrees) |
|---|---|---|
| $[t_{int}, t_1]$ | 0 | $[0 : -5 : -45]$ |
| $[t_1, t_2]$ | $[0 : 5 : 70]$ | −45 |
| $[t_2, t_3]$ | 70 | $[-45 : 5 : 30]$ |
| $[t_3, t_{term}]$ | $[70 : -5 : 50]$ | 30 |

**Table 3. Distribution Information of Uncertain Variables.**

| Variable | Mean | Standard Deviation | Distribution Type |
|---|---|---|---|
| $a_2$ | 475 | 0.475 | Normal |
| $a_3$ | 500 | 0.500 | Normal |
| $a_4$ | 175 | 0.175 | Normal |
| $d_2$ | 300 | 0.300 | Normal |
| $d_6$ | 450 | 0.450 | Normal |
| $\theta_1$ | / | 0.033 | Uniform |
| $\theta_5$ | / | 0.033 | Uniform |

failure probability, the approach roughly matches the benchmark [9]. Table 4 shows the trajectory reliability probability sequence across the interval $[t_{int}, t_{term}]$ with the different threshold(r) assumption. In Table 4, we can see that our technique is more efficient than the others by comparing the number of calls to the performance function.

Table 2 shows the actuation strategy for joints $\theta_1$ *and* $\theta_5$, defining the trajectory for the robotic manipulator across specified time intervals.

Table 3 presents the distribution characteristics of uncertain variables, including their mean, standard deviation, and distribution type, highlighting the normal and uniform distributions used for modeling the parameters in the robotic system.

Results: Table 4 summarizes reliability and error across thresholds, showing that the proposed methods, including Kriging and its variants, perform better than others. The proposed method achieves high accuracy with far fewer evaluations than MCS, whereas MCS required $55 \times 10^4$ number of calls.

Kriging + U and PCE exhibit the highest errors at 2 mm, significantly higher than other methods. For higher thresholds (3 mm and 3.5 mm), errors are maximum, indicating decreased accuracy. MRSM and Kriging + erf show relatively lower error rates across all thresholds (Fig 4). The purple bars (2 mm) dominate the chart, indicating that the error is highest at the 3.5 mm threshold for all methods, and it significantly increases for larger threshold values. Fig 5, The line graph illustrates the number of calls required for different methods at varying threshold values. Each peak and dip highlight the computational effort needed to reach the desired reliability.

**Table 4. Overall Reliability and Error at Different Threshold Values.**

| Method | Reliability (2mm) | Reliability (2.5mm) | Reliability 3mm) | Reliability (3.5mm) |
|---|---|---|---|---|
| MCS | 0.9997 | 0.9972 | 0.9803 | 0.9010 |
| Kriging+U | 0.9999 | 0.9988 | 0.9800 | 0.8789 |
| Kriging+eff | 0.9995 | 0.9970 | 0.9785 | 0.8950 |
| Kriging+erf | 0.9997 | 0.9971 | 0.9791 | 0.8970 |
| MRSM | 0.9992 | 0.9965 | 0.9789 | 0.8932 |
| PCE | 0.9983 | 0.9968 | 0.9708 | 0.8905 |

Error Comparison Analysis: The bar graph presents a comparative analysis of reliability error percentages (%) across different methods for four threshold values (2mm, 2.5mm, 3mm, and 3.5mm). X-axis: Represents different methods (Kriging+U, Kriging+eff, Kriging+erf, MRSM, and PCE). Y-axis: Displays the error percentage (%). Legend: Differentiates the error values for each threshold using distinct colors.

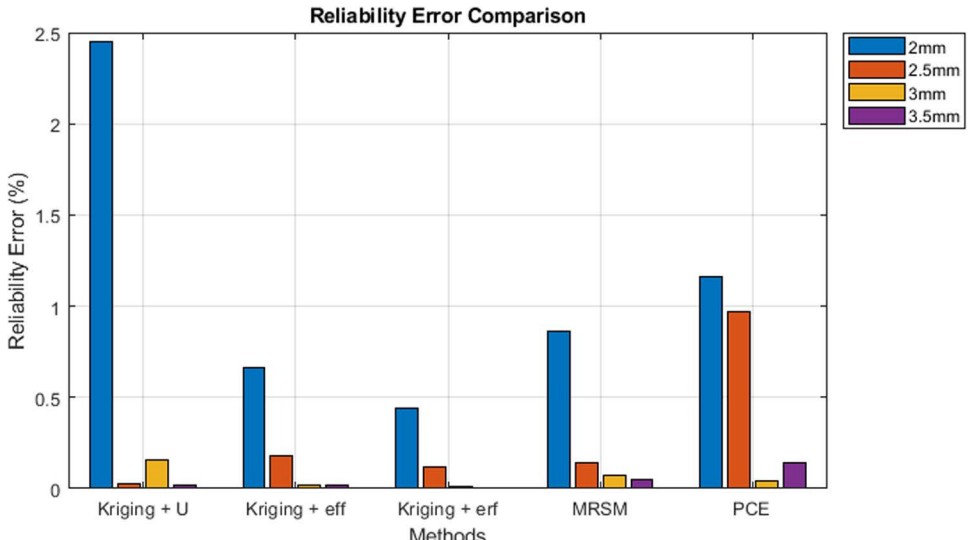

**Fig 4. Accuracy measurement in terms of error.**

PCE consistently requires the highest number of calls across all thresholds, indicating its computational intensity. Kriging+ERF requires the lowest number of calls in all cases, demonstrating its efficiency in achieving results faster. MRSM and Kriging+U show moderate performance, requiring more calls than Kriging+ERF but fewer than PCE. As the threshold increases, the number of calls generally remains stable through methods, except for PCE, which maintains a high value. This visualization effectively compares computational efficiency across reliability estimation techniques, with lower calls indicating higher efficiency in achieving the desired accuracy. Table 5 compares reliability across methods for different time intervals, using the Monte Carlo Simulation (MCS) as a reference. It showcases the robust performance at r=2.5mm of Kriging+erf in the later intervals.

In each period analyzed, with the threshold of 2.5 the distribution of trajectories reveals the performance metrics: from 0 to 1, 543,594 trajectories passed while 6,406 failed out of 550,000 total; from 1 to 2, 535,127 passed and 14,873 failed; from 2 to 3, 537,668 passed and 12,332 failed; and from 3 to 4, 543,411 passed and 6,589 failed. These figures are crucial for calculating the probabilities of failure ($PF1$ to $PF4$) using Equation 38, which typically involves the ratio of failed trajectories to the total trajectories (failed+passed) in each period. Equation 27, often used to calculate reliability, indicates

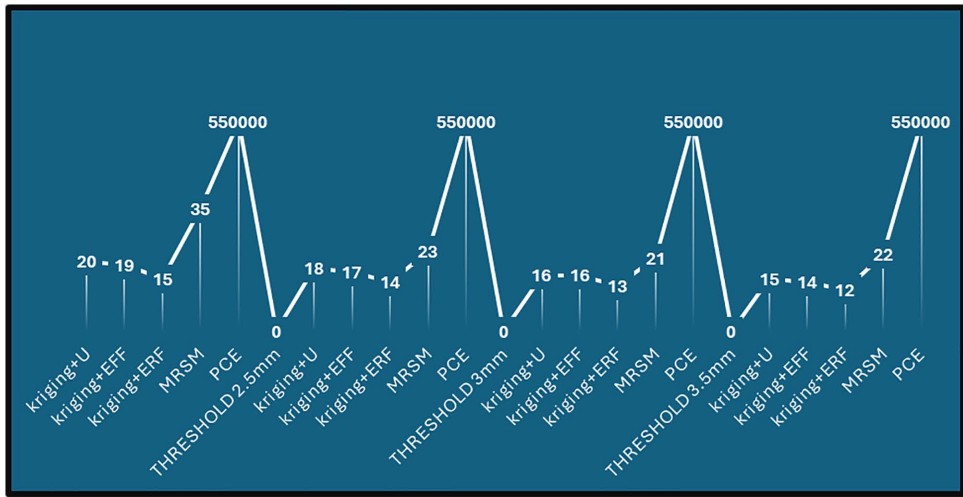

**Fig 5. Number of calls to achieve desired results.**

**Table 5. Reliability ($P_R$) Under $r = 2.5mm$.**

| Method | Reliability $[t_{int},\ t_1]$ | Reliability $[t_1,\ t_2]$ | Reliability $[t_2,\ t_3]$ | Reliability $[t_3,\ t_{term}]$ |
|---|---|---|---|---|
| MCS | 0.9884 | 0.9730 | 0.9776 | 0.9880 |
| Kriging + U | 0.9486 | 0.9606 | 0.9969 | 0.9670 |
| Kriging + eff | 0.9944 | 0.9630 | 0.9959 | 0.9685 |
| Kriging + erf | 0.9857 | 0.9800 | 0.9835 | 0.9884 |
| MRSM | 0.9772 | 0.9835 | 0.9869 | 0.9772 |
| PCE | 0.9849 | 0.9856 | 0.9541 | 0.9568 |

the likelihood that the system will operate without failure over the specified intervals. These formulas give a complete algorithm for evaluating the usage of the trajectory's performance and dependability over some time concurrently with the error percentage of equation 39. In the current study, the same fixed value of 2.5 mm is used in all sample sizes in each of the trajectory intervals; the MCS is $5.5 \times 10^5$ samples to provide comparable reliability and validity of the results over time in each interval. Error analysis: Fig. 6 illustrates reliability errors (%) across four-time intervals for five different methods. It helps compare the stability and accuracy of each method over time.

Kriging + erf shows the lowest error, ensuring stable performance across all intervals. PCE and Kriging + U exhibit the highest errors, particularly in the initial and final intervals. MRSM and Kriging + eff maintain moderate error levels, balancing efficiency and accuracy. Errors decrease in the middle intervals but rise again in the final stage, especially for PCE. This analysis highlights Kriging + erf as the most reliable method, while PCE shows significant error fluctuations over time. Fig 7 shows the number of calls required by different methods to achieve accurate trajectories at 2.5 mm across four-time intervals.

PCE requires the highest calls consistently, making it computationally expensive. Kriging + ERF has the lowest calls, showing high efficiency. MRSM and Kriging + U have moderate call counts, balancing accuracy and efficiency. Call rankings remain stable across intervals, with PCE always being the most resource-intensive method. This analysis helps in selecting the most efficient method for accurate trajectory estimation. This example demonstrates the use of the proposed method for a 6-dof robotic manipulator with an emphasis on efficiency of reliability estimation. All Kriging variants: Kriging + U, eff, erf, show a substantial reduction in function evaluations (from 550,000 calls to 12–20 calls) with

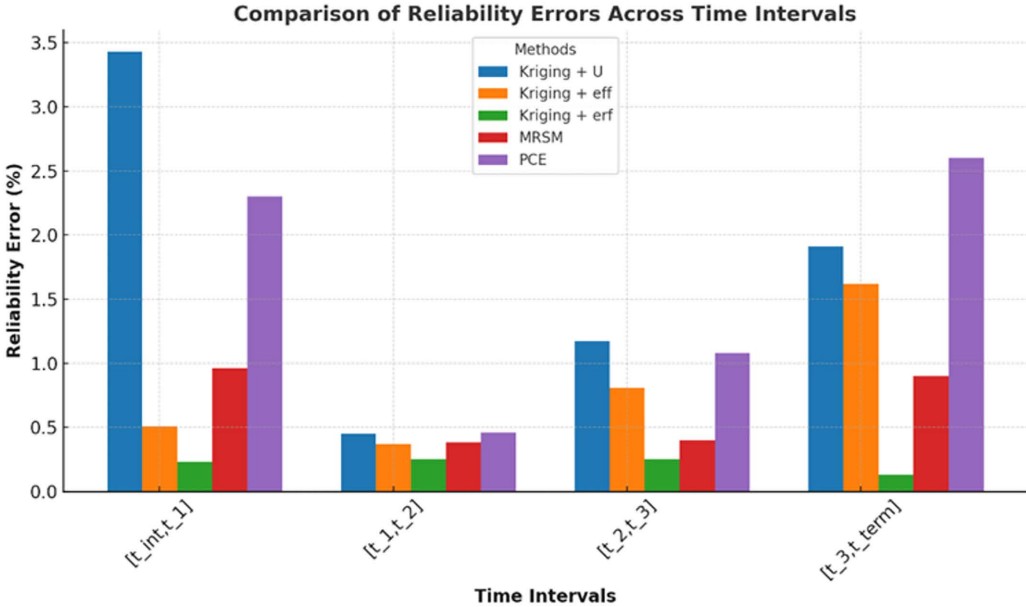

**Fig 6. Reliability error comparison across time intervals for different methods.**

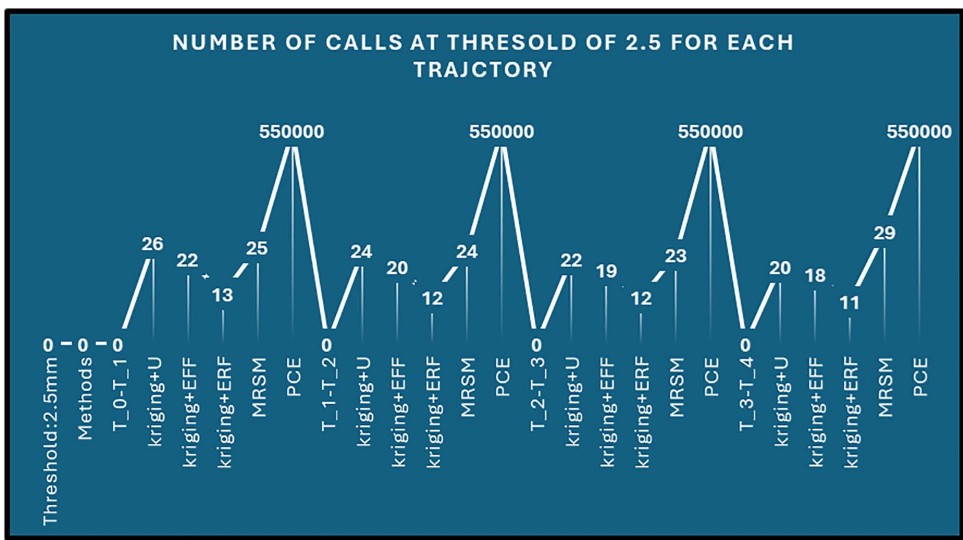

**Fig 7. Number of calls to achieve accurate trajectories.**

reliability levels close to central MCS on the varied threshold ranging from 2 mm to 3.5 mm. A breakdown of the analysis based on the intervals of each of the trajectories and using a 2.5 mm criterion throughout is presented above in Table 5. This approach benchmarked against MCS at $5.5 \times 10^5$ samples, ensures a robust comparison of reliability performance across discrete time intervals: $[t_{int}, t_1]$, $[t_1, t_2]$, $[t_2, t_3]$ and $[t_3, t_{term}]$.. That is why it is critical to emphasize that the approaches suggested in this paper prove the ability of the methods underpinning numerous reliability predictions built into dynamic operative schedules.

## 4.2. Reliability Analysis of the UR-5 Robotic Manipulator

UR5 is one of the products of Universal Robots and it is a smart industrial robot with integrated PLC organized as a collaborative and adaptive robot platform for shared environments. Its payload is 5 kg and reach is 850 mm, and its main applications include repetitive and dangerous operations such as pick and place, palletizing, and quality inspection. Their six DOFs provide great versatility, accuracy, and longevity of the tool, which makes it a favorite among industries [40]. Integration of forward and inverse kinematics derived from MATLAB and employed with Force Torque sensors and different end effectors augment its operation. For reliability, as well as failure probability evaluation, approaches like Monte Carlo Simulation, Kriging + U, Kriging + erf, and Kriging + eff were used. The UR5 workspace and trajectory depicted below are derived from the Peter Corke Robotics Toolbox using Denavit- Hartenberg parameters [21] as described below in Table 10.

Fig 8 showcases the UR5 robotic manipulator following a planned trajectory in 3D space. The overlayed plot illustrates the end-effector's path along $X$, $Y$ and $Z$ axes across various time intervals ($[T_{int} - T_1], [T_1 - T_2], [T_2 - T_3], [T_3 - T_4], [T_4 - T_{term}]$) demonstrating precise adjustments and movement dynamics throughout the trajectory. The following Table 6 presents the Denavit-Hartenberg (DH) parameters for each joint of the robotic manipulator. These parameters are crucial for defining the kinematic relationships that govern the manipulator's motion.

Table 7 summarizes the joint angles ($\theta_1$ and $\theta_5$) for the robotic manipulator across specified time intervals, highlighting the control strategy for precise movement.

Table 8 presents the sampling method for key variables in the robotic manipulator's model. It includes the meaning, standard deviation, and distribution type for each parameter, indicating the statistical approach used to characterize the uncertainties.

The reliability and probability of failure ($P_F$) were calculated for different safe boundaries using the four algorithms. Each method produced different error metrics. The best results were observed at a safe boundary of 2.2 mm. The reliability

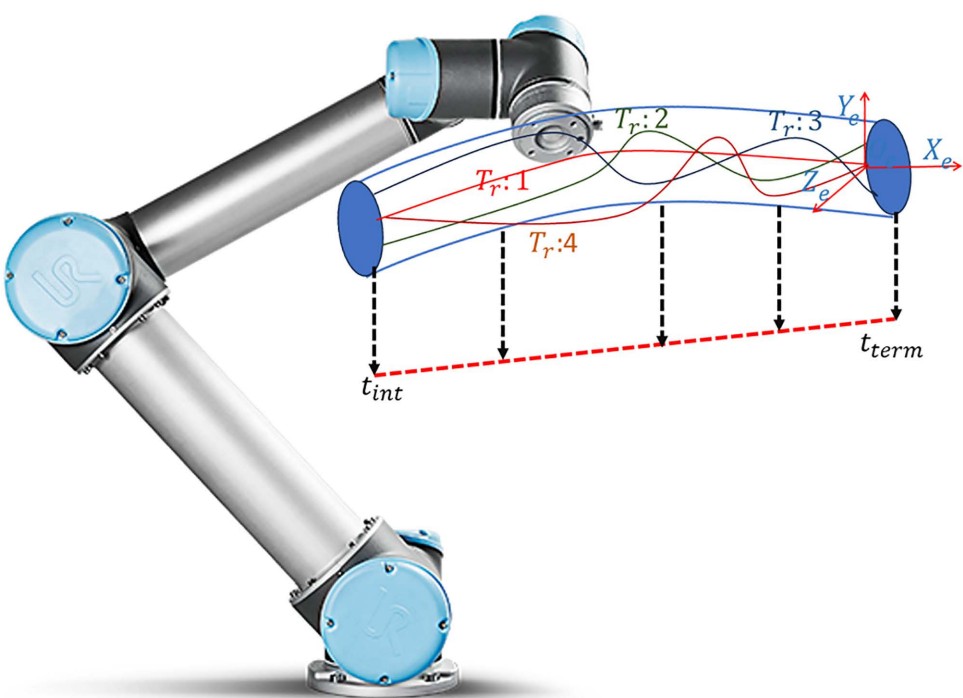

**Fig 8. Workspace for reliable trajectories.**

**Table 6. DH parameters.**

| Joints | $\theta_i$ | $\alpha_{i-1}$ | $a_{i-1}$ (mm) | $d_i$ (mm) |
|---|---|---|---|---|
| 1 | $\theta_1$ | 90 | 0 | 89.2 |
| 2 | −28.49 | 0 | 425 | 0 |
| 3 | 89.82 | 0 | 392 | 0 |
| 4 | 58.67 | 90 | 0 | 109.4 |
| 5 | $\theta_5$ | −90 | 0 | 94.75 |
| 6 | −179.99 | 0 | 0 | 82.5 |

**Table 7. Actuation Mechanism.**

| *Time Interval* | $[t_{int} - t_1]$ | $[t_1 - t_2]$ | $[t_2 - t_3]$ | $[t_3 - t_4]$ | $[t_4 - t_{term}]$ |
|---|---|---|---|---|---|
| $\theta_1$ | 90 | 90 : 20 : 250 | 250 | 250 : 20 : 450 | 450 |
| $\theta_5$ | −90 : 20 : 90 | 90 | 90 : 20 : 170 | 170 | 170 : 20 : 270 |

**Table 8. Sampling method.**

| Variables | Mean | Standard Deviation | Distribution Type |
|---|---|---|---|
| $a_2$ | 425 | 0.4250 | *Normal* |
| $a_3$ | 392 | 0.3920 | *Normal* |
| $d_1$ | 89.2 | 0.0892 | *Normal* |
| $d_4$ | 109.4 | 0.1094 | *Normal* |
| $d_5$ | 94.75 | 0.09475 | *Normal* |
| $d_6$ | 82.5 | 0.0825 | *Normal* |
| $\theta_1$ | $\theta1$ | 0.0183 | *Uniform* |
| $\theta_5$ | $\theta5$ | 0.0183 | *Uniform* |

$P_R(t_{int}, t_{term})$ is calculated. For time-dependent positional reliability, the probability of failure $P_F(t)$ at time $t$ can be calculated as: $P_F(t) = 1 - P_R(t)$ where $P_R(t)$ is the reliability at the time.

## Results

The ideal trajectory of the UR5 robot is shown in Fig 9.

The graphs present five distinct spatial trajectories that the robotic manipulator follows, each demonstrating varied positional paths across the $X$, $Y$, and $Z$ dimensions. This visualization highlights the manipulator's versatility in executing complex movements. Table 9 compares the performance of various methods in terms of reliability ($1.5mm$, $2mm$, $2.5mm$, and $3mm$). The Monte Carlo Simulation (MCS) is taken as a benchmark, the Kriging-based methods and Polynomial Chaos Expansion (PCE) exhibited varying reliability levels with fewer calls, indicating their efficiency in achieving near-optimal reliability with reduced computational effort.

Error analysis: Fig 10 compares reliability errors (%) across four thresholds (1.5 mm, 2 mm, 2.5 mm, and 3 mm) for five methods: Kriging + U, Kriging + eff, Kriging + erf, MRSM, and PCE.

Where Kriging + U shows the highest error at 2.5 mm and 3 mm, Kriging + erf consistently has the lowest error across all thresholds. MRSM's error increases significantly at 3 mm, while PCE remains moderate. Kriging + eff maintains a stable error trend. Errors generally decrease with an increasing threshold, except for MRSM at 3 mm. Fig 11 displays the

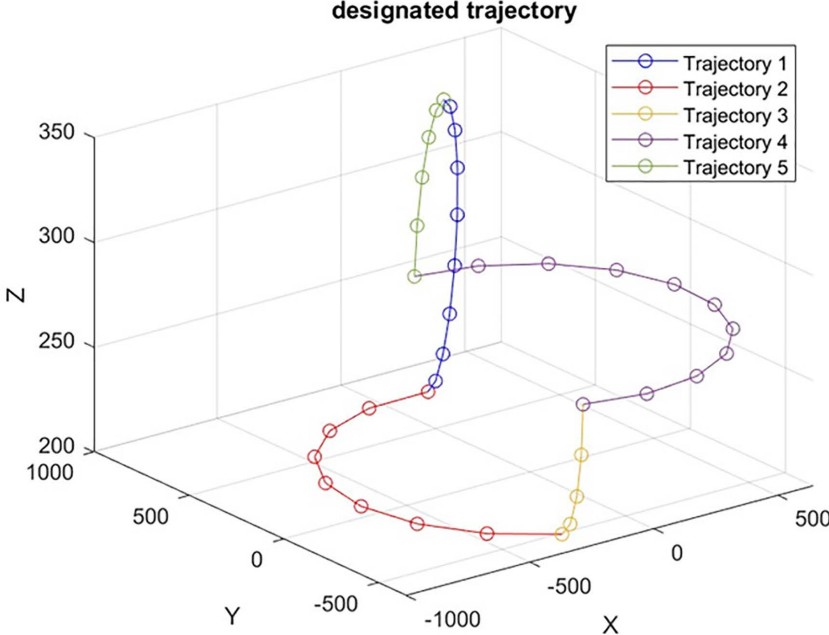

**Fig 9. Ideal Trajectory of the UR5 Robot.**

**Table 9. Overall Reliability and Error at Different Threshold Values.**

| Method | Reliability (1.5 mm) | Reliability (2 mm) | Reliability (2.5 mm) | Reliability (3 mm) |
|---|---|---|---|---|
| MCS | 0.9890 | 0.9803 | 0.9972 | 0.9905 |
| Kriging+U | 0.8410 | 0.9701 | 0.9814 | 0.9912 |
| Kriging+eff | 0.9059 | 0.9583 | 0.9698 | 0.9799 |
| Kriging+erf | 0.9189 | 0.9881 | 0.9898 | 0.9899 |
| MRSM | 0.8821 | 0.9578 | 0.9759 | 0.9797 |
| PCE | 0.8415 | 0.9584 | 0.9595 | 0.9602 |

number of calls required for different threshold values (1.5 mm, 2 mm, 2.5 mm, and 3 mm) across five methods: Kriging+U, Kriging+eff, Kriging+erf, MRSM, and PCE.

Key Observations: PCE consistently requires the highest number of calls, Kriging+erf has the lowest number of calls, indicating computational efficiency, MRSM and Kriging+eff show moderate call values, varying slightly across thresholds. The number of calls generally increases with threshold size, except for Kriging+erf, which remains low and stable. Table 10 showcases the reliability performance of various methods across five critical time intervals, illustrating trajectories from $t_{int}$ to $t_{term}$. Monte Carlo Simulation (MCS) serves as a benchmark. Impressively, the surrogate models demonstrate excellent performance, with reliability values closely aligned with MCS, underscoring their effectiveness. Among these, the Kriging+erf method shines, affirming its strong suitability for real-time applications.

By setting a threshold of $r = 2mm$, In the first period, 3473 trajectories failed out of 550,000 total, while 546527 trajectories passed. Moving to the second period, 3298 trajectories failed, with 546702 trajectories passing. The third period saw 1497 failures and 548503 passes. In the fourth period, 3380 trajectories failed, and 546620 trajectories passed. Finally, in the fifth period, 2598 trajectories failed, while 547402 trajectories passed. Fig 12 threshold of 2 mm, the proposed

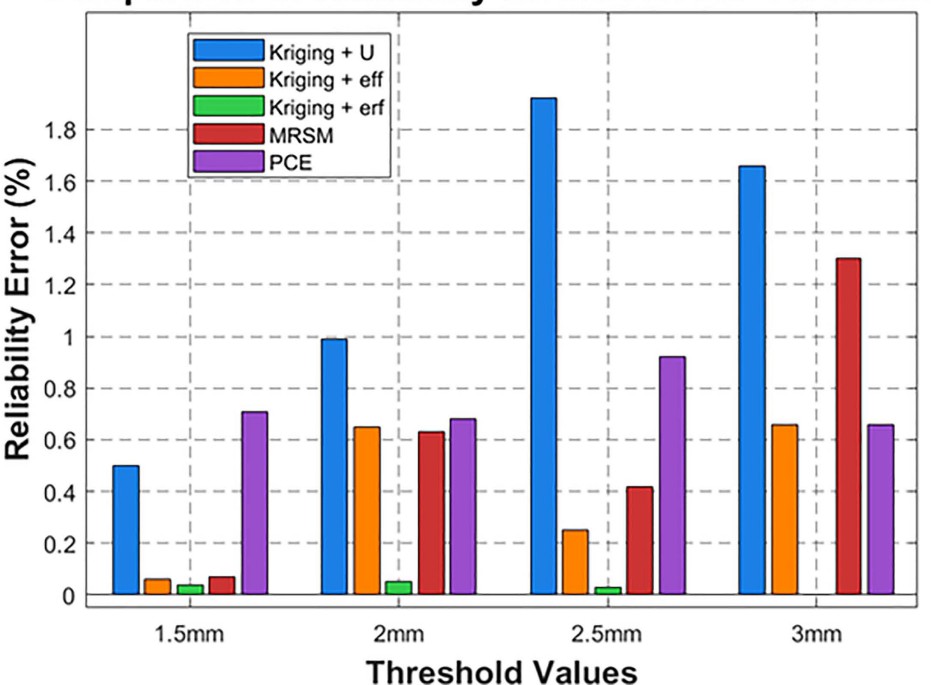

**Fig 10. Error analysis.**

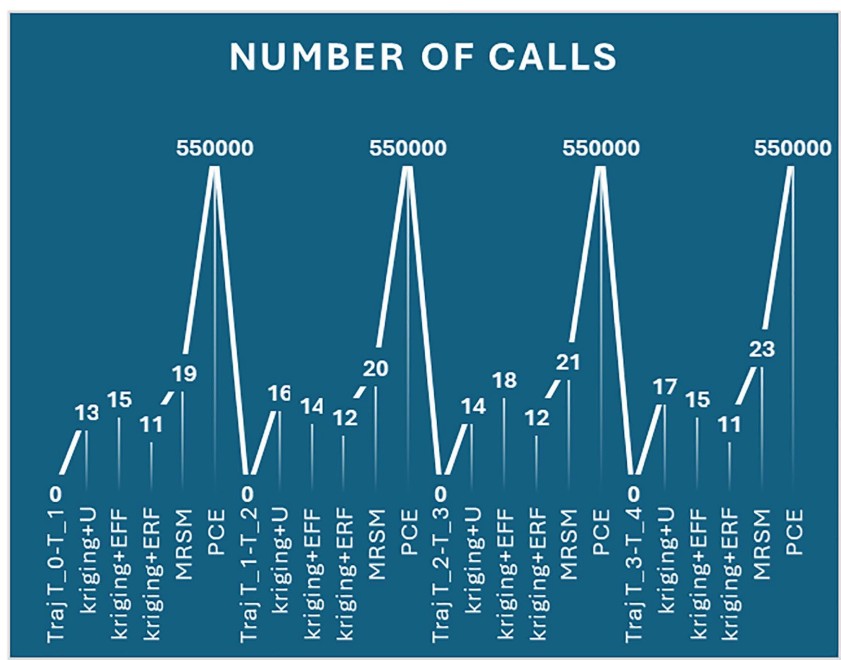

**Fig 11. Number of Calls Required for Different Thresholds Across Methods.**

**Table 10. Reliability ($P_R$) Under $r = 2mm$.**

| Method | Reliability $[t_{int}, t_1]$ | Reliability $[t_1, t_2]$ | Reliability $[t_2, t_3]$ | Reliability $[t_3, t_4]$ | Reliability $[t_4, t_5]$ |
|---|---|---|---|---|---|
| MCS | 0.9937 | 0.9940 | 0.9973 | 0.9939 | 0.9953 |
| Kriging+U | 0.9984 | 0.9985 | 0.9991 | 0.9927 | 0.9959 |
| Kriging+eff | 0.9891 | 0.9953 | 0.9995 | 0.9975 | 0.9945 |
| Kriging+erf | 0.9924 | 0.9942 | 0.9980 | 0.9932 | 0.9954 |
| MRSM | 0.9958 | 0.9976 | 0.9982 | 0.9913 | 0.9952 |
| PCE | 0.9960 | 0.9980 | 0.9985 | 0.9950 | 0.9955 |

methods—Kriging (with variants U, EFF and ERF), demonstrate a substantial reduction in function evaluations for each trajectory segment, where out of them *kriging + ERF* again performs outstanding compared to the MCS benchmark, which maintains $5.5 \times 10^5$ calls.

The kriging method is proposed for its ability to measure errors locally, adaptively refine the grid, and provide high accuracy with minimal samples. To use PCE, actual data labels are necessary, though it handles difficult problems in fewer steps. On the other hand, doing MRSM takes longer since it requires an in-depth study of the covariance structure. Despite this reduction in computational effort, these methods achieve impressively high reliability, with error margins kept to a minimal range. Each approach shows consistent accuracy across different trajectory intervals $[t_{int}, t_1]$, $[t_1, t_2]$, $[t_2, t_3]$, $[t_3, t_4]$ and $[t_4, t_{term}]$ confirming their effectiveness in delivering reliable estimates with far fewer calls. The minimal errors in reliability, typically below 1%, reinforce the precision of these methods, making them efficient and highly accurate alternatives to traditional MCS. Fig 13 illustrates the number of calls required by different methods to achieve an accurate trajectory at a threshold of 2 mm across multiple trials.

The x-axis represents different trials and methods, including Kriging+U, Kriging+Eff, Kriging+Erf, MRSM, and PCE. The y-axis shows the corresponding number of function calls required to reach the accurate trajectory. PCE consistently requires the highest number of calls across all trials. Kriging+Erf requires the least number of calls, indicating its efficiency

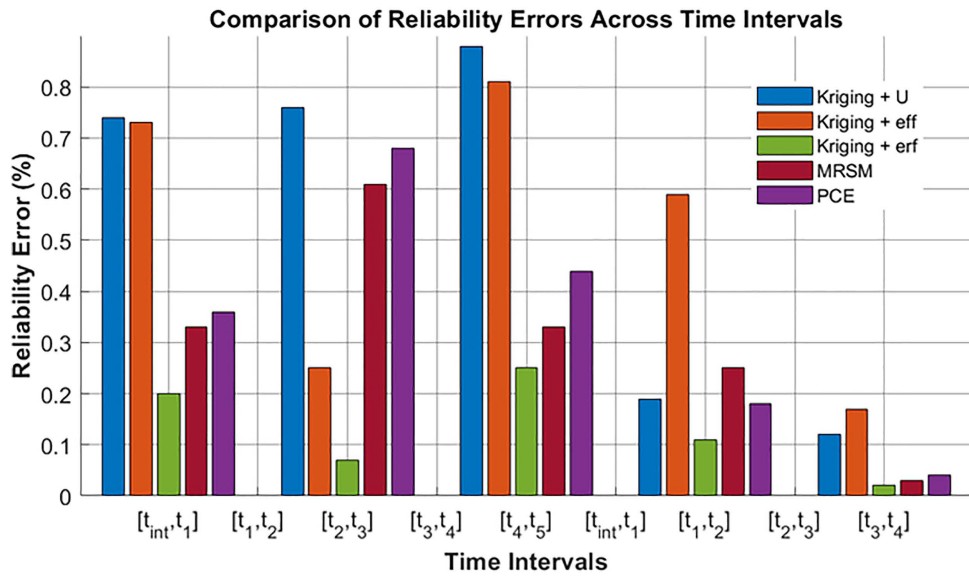

**Fig 12. Error calculation for each trajectory.**

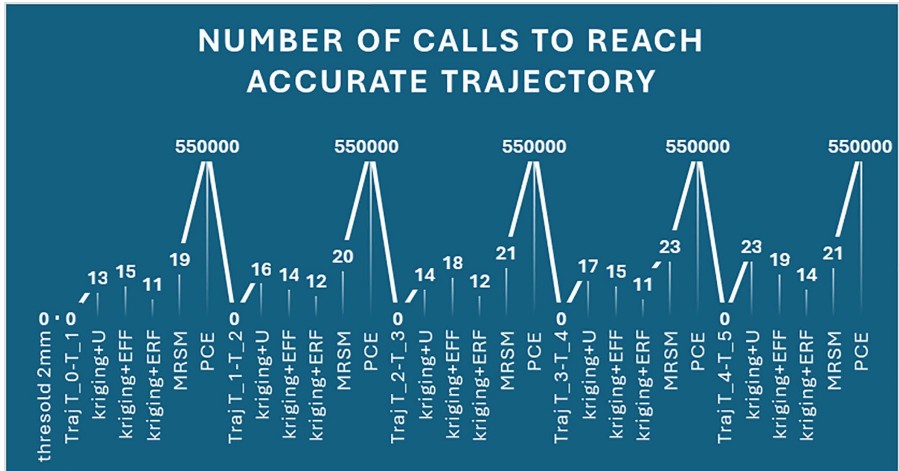

**Fig 13. Number of Calls for UR5 to Reach the accurate Trajectory.**

in achieving accuracy with minimal iterations. MRSM and Kriging + U show moderate call counts, varying across different trials. The peaks in the graph correspond to PCE, while the lower points correspond to more efficient methods. This plot provides insights into the computational efficiency of different methods in reaching accurate trajectory solutions. These results indicate consistent reliability and probability of failure overall time spans for each algorithm. This further emphasizes the proposed method's effectiveness and accuracy in maintaining high positional reliability of the UR5 robotic manipulator throughout various operational intervals.

## 5. Discussion

The discussion section emphasizes the critical insights derived from the study, particularly the importance of time-based reliability assessment for 6DOF manipulators in robotics production. Precision and repeatability are identified as pivotal factors for ensuring highly predictable manipulator performance. To evaluate these aspects, three surrogate algorithms— DACE-fit model Kriging + U, Kriging + EFF, and Kriging + ERF—were employed. These models were selected not only to assess potential improvements in manipulator operation precision but also to provide a reasonable estimate of computational expenses. Among these, the Kriging-based models, particularly Kriging + ERF, demonstrated superior performance in terms of locational precision and computational efficiency. This is a key insight, as Kriging was chosen for its ability to provide sample-wise error estimates, a capability that other algorithms lack. This feature is crucial for achieving high precision and reliability in manipulator performance, especially in dynamic industrial environments. The study also highlights the significance of Gaussian Process sampling in capturing the time-dependent dynamics of manipulator performance. By focusing on actuation sites such as $\theta_1$ and $\theta_5$, the research underscores how changes in actuation points can significantly influence manipulator performance under varying conditions. The mathematical examples presented, which analyzed performance across different time intervals, were designed to replicate real-world scenarios and validate the algorithms within the system's application modes. This approach allowed for a comprehensive comparison of the economy and effectiveness of each algorithm, further reinforcing the superiority of Kriging + ERF in achieving high precision with relatively low computational demands.

As Table 11 shows, Kriging + ERF has the best combination of accuracy and efficiency among the techniques compared. The ERF is introduced as a targeted enhancement over standard Kriging, allowing adaptive sampling in uncertain regions while maintaining low computational cost. Its ability to deliver sample-wise predictions with local MSE makes it especially suitable for time-dependent reliability analysis in robotics. The proposed framework is model-independent and

**Table 11. Method comparison summary.**

| Method | Advantages | Limitations |
|---|---|---|
| MCS | Accurate, robust | Very slow, high computation |
| PCE | Fast | Weak for nonlinear/high-dimensional |
| MRSM | Multi-output, Adaptive, flexible | Complex, slow training |
| Kriging+ERF | Accurate, adaptive, efficient | Needs moderate tuning |

can be extended to robots with varying or redundant DOFs by updating the kinematic parameter set and retraining the surrogate model, without modifying the core algorithm. Moreover, the findings suggest that further mathematical modeling, particularly using enhanced Kriging techniques like ERF, is essential for advancing the precision and reliability of robotic manipulators in position control. These insights not only contribute to the global database on robotic kinematics but also open avenues for future research aimed at improving manipulator efficiency across diverse industrial applications. By focusing on these key insights, the discussion underscores the transformative potential of advanced modeling techniques in robotics, paving the way for more reliable and efficient manipulator systems. Although the current study is based on fixed trajectories, the method is efficient and flexible enough to be applied in dynamic tasks such as robotic inspection, assembly, or human-robot collaboration, where trajectory adjustments are guided by real-time feedback. The UR-5 robot used in this study is a real industrial platform, and its parameters were derived from the official technical manual to ensure practical relevance [41].

## 6. Conclusion

This work aims to revolutionize trajectory optimization in robotic systems by enhancing robotic kinematics and introducing advanced sampling and prediction approaches. The proposed solution focuses on three primary concerns: reducing the computational complexity of robot control, improving the control accuracy of the Universal Robot UR-5, and implementing an adaptive sampling method. Computational expenses are significantly reduced by approximately compared to traditional methods like Monte Carlo Simulation (MCS). Furthermore, the approach accurately forecasts positional reliability with minimal error, offering improved precision. The proposed algorithm performs exceptionally well on the UR5 industrial robotic manipulators, where the accuracy is very high. The results from the surrogate model are very close to those obtained from Monte Carlo Simulation (MCS), highlighting its effectiveness and reliability for industrial applications. This demonstrates the superior efficiency and reliability of Kriging+Erf in practical applications. From both a methodological and practical standpoint, the proposed approach is robust and shows considerable potential to enhance the accuracy and efficiency of robotic systems in industrial automation and other technological, engineering, or scientific applications. The reliability estimation framework developed here serves as a foundation for further research and improvements, positioning this study as a valuable contribution to the field of robotic kinematics. Also, using the proposed methodology, there are no structural barriers to handling larger or more detailed types of input data. In this work, $\theta_1$ and $\theta_5$ were used as actuation variables to define the trajectory. However, the same framework can easily integrate additional actuation mechanisms or extended parameter spaces, demonstrating its robustness for handling more sophisticated manipulator architectures while preserving computational efficiency and predictive accuracy.

## Author contributions

**Conceptualization:** Keenjhar Ayoob, Hassan Elahi, Tayyab Zafar, Amir Hamza, Zhonglai Wang.

**Data curation:** Keenjhar Ayoob.

**Formal analysis:** Tayyab Zafar, Amir Hamza.

**Methodology:** Keenjhar Ayoob.

**Software:** Keenjhar Ayoob.

**Supervision:** Hassan Elahi, Tayyab Zafar.

**Validation:** Keenjhar Ayoob, Hassan Elahi, Tayyab Zafar.

**Writing – original draft:** Keenjhar Ayoob.

**Writing – review & editing:** Keenjhar Ayoob, Tayyab Zafar, Amir Hamza.

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
