## [Decision Letter · Decision Letter 0]

19 May 2025

PONE-D-25-09414Surrogate Modeling for Time-Dependent Reliability Analysis of Robotic Manipulator TrajectoriesPLOS ONE

Dear Dr. Elahi,

Thank you for submitting your manuscript to PLOS ONE. After careful consideration, we feel that it has merit but does not fully meet PLOS ONE’s publication criteria as it currently stands. Therefore, we invite you to submit a revised version of the manuscript that addresses the points raised during the review process.

We look forward to receiving your revised manuscript.

Kind regards,

Van Thanh Tien Nguyen, Ph.D.

Academic Editor

PLOS ONE

Journal Requirements:

Reviewers' comments:

Reviewer's Responses to Questions

**Comments to the Author**

1. Is the manuscript technically sound, and do the data support the conclusions?

Reviewer #1: Yes

Reviewer #2: Yes

Reviewer #3: Yes

2. Has the statistical analysis been performed appropriately and rigorously? 

Reviewer #1: N/A

Reviewer #2: Yes

Reviewer #3: Yes

3. Have the authors made all data underlying the findings in their manuscript fully available?

Reviewer #1: Yes

Reviewer #2: Yes

Reviewer #3: Yes

4. Is the manuscript presented in an intelligible fashion and written in standard English?

Reviewer #1: Yes

Reviewer #2: Yes

Reviewer #3: Yes

5. Review Comments to the Author

Reviewer #1: It would be helpful to briefly mention any potential challenges or areas where the proposed methodology may need further refinement, especially as it relates to scaling up to larger or more complex systems.

A concise summary of the model's performance in comparison to alternative approaches would help reinforce the importance of the proposed methodology in the conclusion.

Reviewer #2: The paper is very welll written and organised. The results presnted in the paper is very promising. Thereferences used in this paper is relevent and sufficient. Overall good paper and good contribution in the field of robotics.

The approach reduces computational complexity while maintaining prediction accuracy. Compared to Monte Carlo Simulation (MCS), the proposed Krigingbased method reduces the number of function evaluations by over 98%. The proposed method is validated on two 6-DOF industrial robots and shown improved computational efficiency and accuracy.

Reviewer #3: This paper explores a Kriging surrogate-based method for analyzing the time-dependent reliability of robotic manipulator trajectories. It aims to address the limitations of traditional methods in handling nonlinear performance functions and trajectory reliability. Before acceptance, the following issues need to be resolved:

1. In section “3. Proposed Methodology”, the Kriging surrogate model is said to cut computation complexity while keeping prediction accuracy. Could you clarify how the initial sample points are chosen in the Kriging model? How do the quantity and location of these points affect the final result's accuracy?

2. The paper notes prediction errors in the Kriging model, assessed by ERF and MSE. But the analysis of error sources and propagation mechanisms is insufficient.

3. In section “4. Case Studies and Results, comparisons are made between the proposed method and existing ones like Monte Carlo Simulation, MRSM, and PCE. Yet, analyzing each method's advantages and disadvantages is rather general.

4. The method's effectiveness was verified on two 6-DOF industrial robots. However, in real-world industrial settings, there are robots of diverse configurations and DOFs. How can the method be extended to more complex robot systems, such as redundant DOF or specially structured robots?

5. The proposed method mainly analyzes reliability under fixed trajectories. However, in dynamic real-world settings, robots may need to adjust trajectories based on real-time feedback.

6. PLOS authors have the option to publish the peer review history of their article (what does this mean? ). If published, this will include your full peer review and any attached files.

**Do you want your identity to be public for this peer review?** For information about this choice, including consent withdrawal, please see our Privacy Policy .

Reviewer #1: No

Reviewer #2: **Yes: ** Prof. (Dr.) Vijay Kumar Banga

Reviewer #3: No

---

## [Author Response · Author response to Decision Letter 1]

29 May 2025

Response to Reviewers

Comments Addressed

It would be helpful to briefly mention any potential challenges or areas where the proposed methodology may need further refinement, especially as it relates to scaling up to larger or more complex systems. Response:

We thank the reviewer for this important observation. To address this, we have clarified in the Conclusion section that the proposed methodology is structurally flexible and can be extended to handle larger or more complex systems. Specifically, we added the following sentence:

“Also, using the proposed methodology, there are no structural barriers to handling larger or more detailed types of input data. In this work, θ₁ and θ₅ were used as actuation variables to define the trajectory. However, the same framework can easily integrate additional actuation mechanisms or extended parameter spaces, demonstrating its robustness for handling more sophisticated manipulator architectures while preserving computational efficiency and predictive accuracy.”

Reviewer#1

Reviewer#2

Comments Addressed

The paper is very welll written and organised. The results presnted in the paper is very promising. Thereferences used in this paper is relevent and sufficient. Overall good paper and good contribution in the field of robotics.

The approach reduces computational complexity while maintaining prediction accuracy. Compared to Monte Carlo Simulation (MCS), the proposed Krigingbased method reduces the number of function evaluations by over 98%. The proposed method is validated on two 6-DOF industrial robots and shown improved computational efficiency and accuracy.

We sincerely thank the reviewer for their positive and encouraging feedback. We are grateful for the recognition of the clarity, organization, and relevance of our work, as well as the significance of the proposed methodology. Your appreciation of the computational efficiency and accuracy of our Kriging-based approach, validated on two 6-DOF industrial robots, is highly valued. We are pleased that the contribution is considered valuable in the field of robotics.

Reviewer#3:

This paper explores a Kriging surrogate-based method for analyzing the time-dependent reliability of robotic manipulator trajectories. It aims to address the limitations of traditional methods in handling nonlinear performance functions and trajectory reliability. Before acceptance, the following issues need to be resolved:

In section “3. Proposed Methodology”, the Kriging surrogate model is said to cut computation complexity while keeping prediction accuracy. Could you clarify how the initial sample points are chosen in the Kriging model? How do the quantity and location of these points affect the final result's accuracy?

Response:

We thank the reviewer for this helpful comment. To address it, we have added a detailed explanation in Section 3 (Proposed Methodology), immediately after Equation (16), describing how the initial sample points are selected and how their distribution impacts model accuracy. The following text and pseudocode were inserted:

“The output matrix Ŷ in Equation (16) is computed using forward kinematics from a set of 10 input samples. These samples are generated by perturbing the Denavit–Hartenberg (DH) parameters based on realistic uncertainty models. Joint angles, θ_1 and θ_5 are calculated from uniform distributions, while link lengths and offsets follow normal distributions. It provides an even distribution of points throughout the space of given inputs. Forward kinematics (using the Peter Corke Robotics Toolbox) is used to compute corresponding end-effector positions. The process also makes the error rate more precise by adjusting the model in problem areas.

Pseudocode: Initial Sample Generation

for i = 1:N_samples

θ_1 = unifrnd(90,450);

θ_5 = unifrnd(-90,270);

d = normrnd(d_nominal,σ_d);

a = normrnd(a_nominal,σ_a);

α = fixed or DH-defined;

T = Forward_Kinematics(θ,d,a,α);

Store T and DH parameters;

End

“

The paper notes prediction errors in the Kriging model, assessed by ERF and MSE. But the analysis of error sources and propagation mechanisms is insufficient.

Response:

We thank the reviewer for pointing this out. We have clarified how the Kriging surrogate model inherently quantifies sample-wise prediction uncertainty and how these uncertainties guide adaptive refinement. We also added a comparative note discussing the limitations of MRSM and PCE in this context. These additions better explain the error origin and how the model manages uncertainty propagation.

Changes Made:

Location: Section 3 – Kriging Surrogate Model (after Equations 21 and 22)

Added the following paragraph to clarify the source and handling of prediction uncertainty:

“It is particularly useful that the Kriging surrogate model can generate estimates of the end-effector position and a sample-wise calculation of how uncertain each prediction is. The model, for every input sample, returns the predicted position as well as a Mean Squared Error reading that shows how confident it was in that prediction. This makes the model particularly powerful for reliability analysis, as it allows error-sensitive decisions to be made at each test point. The adaptive sampling loop counts on sample-wise MSE values to direct additional training to the regions that are still uncertain for the model. This built-in capability to quantify and localize error propagation enhances both model transparency and reliability assessment accuracy.”

Location: Discussion Section (after analysis of Figure 13)

Added the following short paragraph to compare MRSM and PCE limitations:

“The kriging method is proposed for its ability to measure errors locally, adaptively refine the grid, and provide high accuracy with a minimal number of samples. To use PCE, actual data labels are necessary, though it handles difficult problems in fewer steps. On the other hand, doing MRSM takes longer since it requires an in-depth study of the covariance structure.”

In section “4. Case Studies and Results, comparisons are made between the proposed method and existing ones like Monte Carlo Simulation, MRSM, and PCE. Yet, analyzing each method's advantages and disadvantages is rather general.

Response:

Thank you for the comment. To address this, we have added a concise comparative table (Table 11) and the following paragraph in the Discussion section after the sentence ending with “low computational demands”:

“As Table 11 shows, Kriging + ERF has the best combination of accuracy and efficiency among the techniques compared. The ERF is introduced as a targeted enhancement over standard Kriging, allowing adaptive sampling in uncertain regions while maintaining low computational cost. Its ability to deliver sample-wise predictions with local MSE makes it especially suitable for time-dependent reliability analysis in robotics.”

The method's effectiveness was verified on two 6-DOF industrial robots. However, in real-world industrial settings, there are robots of diverse configurations and DOFs. How can the method be extended to more complex robot systems, such as redundant DOF or specially structured robots? Response:

Thank you for the insightful comment. We have clarified this point in the Discussion section by adding the following sentence after the justification of Kriging + ERF:

“The proposed framework is model-independent and can be extended to robots with varying or redundant DOFs by updating the kinematic parameter set and retraining the surrogate model, without modifying the core algorithm.”

The proposed method mainly analyzes reliability under fixed trajectories. However, in dynamic real-world settings, robots may need to adjust trajectories based on real-time feedback. Response:

Thank you for the valuable comment. While our current analysis focuses on fixed trajectories, we have clarified in the Discussion section that the method is computationally efficient and flexible enough to be applied in dynamic applications such as inspection, assembly, or human-robot collaboration. We also noted that the UR-5 robot used in the study is a real industrial platform, and its configuration parameters were derived from the official user manual to ensure practical relevance. The following sentence was added at the end of the Discussion section:

“Although the current study is based on fixed trajectories, the method is efficient and flexible enough to be applied in dynamic tasks such as robotic inspection, assembly, or human-robot collaboration, where trajectory adjustments are guided by real-time feedback. The UR-5 robot used in this study is a real industrial platform, and its parameters were derived from the official technical manual to ensure practical relevance.”

---

## [Decision Letter · Decision Letter 1]

18 Aug 2025

Surrogate Modeling for Time-Dependent Reliability Analysis of Robotic Manipulator Trajectories

PONE-D-25-09414R1

Dear Dr. Elahi,

We’re pleased to inform you that your manuscript has been judged scientifically suitable for publication and will be formally accepted for publication once it meets all outstanding technical requirements.

Kind regards,

Abbasali Sadeghi, Ph.D.

Academic Editor

PLOS ONE

Additional Editor Comments (optional):

Dear Authors;

Thank you for revisions.

Reviewers' comments:

Reviewer's Responses to Questions

**Comments to the Author**

1. If the authors have adequately addressed your comments raised in a previous round of review and you feel that this manuscript is now acceptable for publication, you may indicate that here to bypass the “Comments to the Author” section, enter your conflict of interest statement in the “Confidential to Editor” section, and submit your "Accept" recommendation.

Reviewer #1: All comments have been addressed

Reviewer #3: All comments have been addressed

2. Is the manuscript technically sound, and do the data support the conclusions?

Reviewer #1: Yes

Reviewer #3: Yes

3. Has the statistical analysis been performed appropriately and rigorously? 

Reviewer #1: Yes

Reviewer #3: Yes

4. Have the authors made all data underlying the findings in their manuscript fully available?

Reviewer #1: Yes

Reviewer #3: Yes

5. Is the manuscript presented in an intelligible fashion and written in standard English?

Reviewer #1: (No Response)

Reviewer #3: Yes

6. Review Comments to the Author

Reviewer #1: The authors have addressed all comments.

Before the paper's publication, it is recommended the inclusion of:

1. In the introduction, the authors should add a paragraph with a literature revision about nonprobabilistic methods to assess the reliability of robotic manipulator trajectories should be added. The following references are recommended:

https://doi.org/10.3390/robotics13110155

https://doi.org/10.3390/machines10100917

https://doi.org/10.1109/TR.2023.3292089

https://doi.org/10.1017/S0263574721000187

2. The limitations of the present approach should be clearly discussed in section "5. Discussion": limitation of the numerical method and also application to experimental case study.

Reviewer #3: (No Response)

7. PLOS authors have the option to publish the peer review history of their article (what does this mean? ). If published, this will include your full peer review and any attached files.

**Do you want your identity to be public for this peer review?** For information about this choice, including consent withdrawal, please see our Privacy Policy .

Reviewer #1: No

Reviewer #3: No

---

## [Editor Report · Acceptance letter]

PONE-D-25-09414R1

PLOS ONE

Dear Dr. Elahi,

I'm pleased to inform you that your manuscript has been deemed suitable for publication in PLOS ONE. Congratulations! Your manuscript is now being handed over to our production team.

Kind regards,

on behalf of

Dr. Abbasali Sadeghi

Academic Editor

PLOS ONE